# Bio-Synthesized Nanoparticles in Developing Plant Abiotic Stress Resilience: A New Boon for Sustainable Approach

**DOI:** 10.3390/ijms23084452

**Published:** 2022-04-18

**Authors:** Sarika Kumari, Risheek Rahul Khanna, Faroza Nazir, Mohammed Albaqami, Himanshu Chhillar, Iram Wahid, M. Iqbal R. Khan

**Affiliations:** 1Department of Botany, Jamia Hamdard University, New Delhi 110062, India; sarikakumari_sch@jamiahamdard.ac.in (S.K.); khannarisheek@gmail.com (R.R.K.); farozanazir_sch@jamiahamdard.ac.in (F.N.); himanshuchhillar9@gmail.com (H.C.); 2Department of Biology, Faculty of Applied Science, Umm Al-Qura University, Makkah 21955, Saudi Arabia; mmbaqami@uqu.edu.sa; 3Department of Biosciences, Integral University, Lucknow 226026, India; iramicro@gmail.com

**Keywords:** abiotic stress, green synthesized nanoparticles, phytohormones

## Abstract

Agriculture crop development and production may be hampered in the modern era because of the increasing prevalence of ecological problems around the world. In the last few centuries, plant and agrarian scientific experts have shown significant progress in promoting efficient and eco-friendly approaches for the green synthesis of nanoparticles (NPs), which are noteworthy due to their unique physio-biochemical features as well as their possible role and applications. They are thought to be powerful sensing molecules that regulate a wide range of significant physiological and biochemical processes in plants, from germination to senescence, as well as unique strategies for coping with changing environmental circumstances. This review highlights current knowledge on the plant extract-mediated synthesis of NPs, as well as their significance in reprogramming plant traits and ameliorating abiotic stresses. Nano particles-mediated modulation of phytohormone content in response to abiotic stress is also displayed. Additionally, the applications and limitations of green synthesized NPs in various scientific regimes have also been highlighted.

## 1. Introduction

Climate change and growing populations have a significant impact on global agricultural food security [1]. Abiotic stress is one of the most serious global environmental issues, and it has compelled researchers to devote their efforts to ensuring the long-term stability of the ecological system. Drought, salinity, heavy metals, and extreme high and low temperatures are the primary abiotic stresses impacting agricultural production globally [2,3]. These abiotic stresses are coupled to osmotic stress which disrupts ion allocation and plant metabolism. Furthermore, the deterioration of agricultural soils also endangers the health of humans and wildlife [4]. According to the statistical approximation, about 90 percent of agricultural land areas are primarily altered by these stresses [5] resulting in a 70 percent reduction in the production of important food crops [6]. These concerning situations have arisen because of unpredicted climatic variations, anthropogenic activities, and poor agricultural techniques [7]. Abiotic stress circumstances can result in the production of reactive oxygen species (ROS), which, if not detoxified, will undoubtedly minimize soil quality and fertility and impede many cellular functions at varying levels of metabolism, including photosynthesis rate, biochemical changes, carbon assimilation, and membrane permeability, resulting in a reduction in crop production [8,9]. A variety of agrarian and physiological practices is used to mitigate the negative effects of abiotic stress and to promote plant stress adaptability. In recent years, nanotechnology has emerged as a promising platform in the epoch of agriculture, garnering the attention of researchers from a wide range of sectors, especially in the development of efficient and eco-friendly methodologies for the green synthesis of nanoparticles (NPs), which holds great potential for resolving issues attributed with abiotic stresses to ensure agricultural sustainability [10,11]. With their unique physio-chemical properties, such as remarkable stability within cells, extremely tiny size (1–100 nm), more surface area and reactivity, NPs are gaining enormous significance in the field of molecular biology research [12]. It is well understood that doses of NPs can have both positive and damaging biological effects. At high doses, they cause oxidative damage to biomolecules, which can result in cell death in plants. Nonetheless, at low nanomolar concentrations, NPs act as a key regulator in managing plant growth and development [13]. The application of NPs strengthens plant stress resilience by boosting the radical detoxifying capacity and antioxidant enzymatic activities [14], which significantly aid in regulating the physio-biochemical and metabolic processes in plants [15] (Figure 1). In addition, NPs have a well-known role in plant responses to environmental variables such as heavy metals [16], drought [17,18], salinity [7] and heat stress [19]. They have shown to have significant impact on plant stress responses, largely by serving as a mediator of physiologically and/or environmentally monitored up-regulation of tolerance genes and proteins that link the biochemical pathways and contribute to stress tolerance management. This review reported the synthesis of a diverse range of NPs such as palladium (Pd), iron (Fe), platinum (Pt), gold (Au), silver (Ag), copper (Cu), zinc (Zn), and selenium (Se) using specific plant parts such as leaves, stems, roots, peels, bark, flowers, fruits and seeds. An attempt has been made to address the role of green synthesized NPs in reprogramming plant characteristics under stress-free and stressful environmental circumstances. The present review emphasizes NPs crosstalk with other plant hormones and their role in mitigating abiotic stresses such as salinity, drought, heat and heavy metal.

## 2. Plant Extract Mediated Synthesis of Nanoparticles

Nowadays, green nanotechnology/phyto-nanotechnology has expanded to be an emerging and developing scientific discipline in all areas. It is playing a significant role in facilitating environmentally friendly approaches using green synthesized NPs, which have received considerable attention in developing abiotic stress resilience in plants [20,21]. The use of green synthesized NPs also reduces the labor impact on the ecosystem and promotes the use of environment-benign reagents [22]. Plants have the ability to synthesize a wide array of NPs such as Pd, Fe, Pt, Au, Ag, Cu, Zn, and Se in different plant parts such as leaves, stems, roots, peels, bark, flowers, fruits, and seeds (Table 1). Phytochemicals such as flavonoids, alkaloids, steroids, phenolics and saponins aid in the phytogenic synthesis of NPs by acting as capping, reducing, and stabilizing agents [23]. In this subsection, biogenic synthesized NPs and their potential applications in different fields have been discussed and summarized (Figure 2 and Table 1).

### 2.1. Roots

Root phenology and structural dynamics serve as the substantial plant part that maintains the resource partitioning, biogeochemical processes, spatio-temporal heterogeneity of soil complexes, root-microbe interactions, nutrient availability, and acquisition [41]. All these multidimensional traits and medicinal properties of roots have impacted on their efficacious role in green nanotechnology. Various intrinsic metabolites were reported to act as the reducing as well as the stabilizing agents in the formation of desired NPs. Velmurugan et al. [24] showed that the reducing ability of oxalic acid, ascorbic acid, phenylpropanoids, zingerone, gingerol, shagaols, and paradol present in *Zingiber officinale* root extract aided in the formation of silver NPs (AgNPs) and gold NPs (AuNPs). Similarly, titanium dioxide NPs (TiO_2_NPs) were synthesized from the root extract of *Withania somnifera*, which contains bioactive metabolites including withanolides, sitoindosides, amino acids and flavonoids. These bio-fabricated TiO_2_NPs were determined using energy-dispersive X-ray spectral (EDS) analysis [25]. Saheb et al. [26] examined the hexagonal/rhombohedral lattice system of bio-prepared AgNPs from *Rheum turkestanicum* root extract via X-ray diffraction (XRD). It was found that the high concentrations of phenolic and anthraquinone compounds were probably involved in the reduction insilver ions (Ag^+^) to Ag°, and hence resulted in the formation of AgNPs. In another study, the Fourier transform infrared (FTIR) spectroscopy confirmed the presence of mimosine (β-3-hydroxy-4 pyridone amino acid) in the aqueous root extract of *Mimosa pudica* [42]. This compound reduced the ferrous sulfate (FeSO_4_) to iron oxide NPs (Fe_3_O_4_NPs), thus indicating its potential role as both reducing and stabilizing agents in NPs synthesis. Furthermore, the biomedical applications of root-mediated synthesized NPs have also been documented in the literature. For instance, ZnNPs and AgNPs synthesized from the root extract of *Panax ginseng* [43] and *Glycyrrhiza glabra* [44] showed a significant role in the diagnosis of diseases, particularly cancer and gastric ulcers, respectively.

### 2.2. Leaf

Leaf extracts are considered an excellent source for NPs synthesis. Various parameters such as temperature, pH, reactant concentrations, and reaction time also critically determine the rate of NPs formation [45]. For instance, synthesized platinum NPs (PtNPs) using *Diospyros kaki* leaves were attained with a reaction temperature of 95 °C and a leaf broth concentration of >10%, which played a positive role in controlling the size of synthesized NPs in the range of 2–12 nm. In addition, the FTIR analyses revealed the presence of several functional groups, such as amines, alcohols, ketones, aldehyde and carboxylic acid surrounding the PtNPs. This implies that synthesized PtNPs was an enzyme-independent process, as the rate of PtNPs synthesis was greatest at temperatures as high as 95 °C and there are no peaks coupled with proteins and/or enzymes on FTIR analysis [46]. The bimetallic (Ag-Cu) NPs synthesized using *Vitex negundo* leaf extract were found to exhibit high tensile strength and thermal stability examined through the Universal testing machine (UTM) and thermogravimetric analyzer (TGA), which indicated that these NPs could be used in food packaging [47]. Pandian et al. [29] assessed the adsorption capacity of *Ocimum sanctum* leaf extract using the Langmuir, Freundlich, and Dubinin-Radushkevich isotherm models. In this study, nickel oxide NPs (NiONPs) were found to be efficient in the removal of anionic pollutants from aqueous solution, suggesting their potential role in the environmental remediation. Apart from these metallic NPs, the use of rare earth metals has been widely exploited for the synthesis of NPs; for instance, lanthanum oxide NPs (La_2_O_3_NP) and neodymium oxide NPs (Nd_2_O_3_NP) were synthesized using the leaf extracts of *Eucalyptus globulus* [30] and *Andrographis paniculata* [48] respectively.

### 2.3. Stem

Stem is considered as an eminent source for the biogenic synthesis of NPs. Silver NPs have been extensively synthesized using the stem extract of plants such as *Salvadora persica* [49], *Momordica charantia* [50], and *Coleus aromaticus* [51]. However, the stem-mediated syntheses of other metallic or bimetallic NPs have also been reported in the literature. Venkateswarlu et al. [52] showed the core–shell structure of Fe_3_O_4_-AgNPs synthesized from *Vitis vinifera* stem extract using high-resolution transmission electron microscopy (HR-TEM) studies. These Ag-doped Fe_3_O_4_NPs were found to be reusable due to their excellent magnetic property. Kirupagaran et al. [27] synthesized spherical-shaped selenium NPs (SeNPs) using *Leucas lavandulifolia* stem extract. In this study, the reduction of Se ions to SeNPs was mediated by the phytoconstituents such as polyphenols and water-soluble heterocyclic components, confirmed by the color change due to surface plasmon resonance (SPR) phenomena. Jayappa et al. [28] reported the inhibitory activity of zinc oxide NPs (ZnONPs), synthesized using *Mussaenda frondosa* stem extract, on the α-amylase and α-glucosidase enzymatic activities, indicating their role in treating diabetes mellitus. Similarly, *Swertia chirayita* stem extract was used in ZnONPs synthesis and was found to be potent in terms of enhanced structural, high photocatalytic and antimicrobial activity [53].

### 2.4. Bud

Bud extract possesses diverse compounds such as polyphenols and flavonol glycosides, which are reported to act as oxidizing/reducing or as bio-templates, and are favorable for the synthesis of NPs. Evidence suggests that NPs such as AuNPs [54], AgNPs [55], palladium NPs (PdNPs) [56], copper NPs (CuNPs) [57], and copper oxide NPs (CuONPs) [58] are extensively being synthesized from *Syzygium aromaticum* buds. This is possible due to the presence of a wide array of pharmacologically active chemical constituents such as hydroxybenzoic acids, hydroxyphenyl propens, eugenol, gallic acid derivatives, quercetin, kaempferol, and ferulic acid [59]. Other phytochemicals such as phenolic compounds, flavonoids, and proteins present in *Couropita guianensis* bud extract were used in the synthesis of AgNPs [38]. Similar results were demonstrated by Nima and Ganesan, [60] using the floral bud broth of *C. guianensis*. Lee et al. [31] used *Tussilago farfara* bud extract containing sesquiterpenoid compounds as the reducing agent for the synthesis of spherical shaped AgNPs and AuNPs. In another study, Rawani et al. [32] confirmed the synthesis of AgNPs from *Polianthus tuberosa* bud extract and revealed the presence of functional groups such as amide, phosphate, ketones, alkenes, nitro, aromatic, and alkyl groups through FTIR analysis of synthesized AgNPs.

### 2.5. Flower

Flowers contain various phytochemicals such as flavonoids, anthocyanins, carotenoids and phenolics that possess pharmacological properties. Using simple extraction protocols, the phytochemicals in the flowers can be extracted. This approach is comparatively inexpensive and eco-friendly, justifying the reason for the tremendous use of flower extracts in the synthesis of NPs [61]. A generalized mechanism for NPs’ synthesis of flower extract is suggested by Kumar et al. [62]. It involves the addition of metal salts to the flower extract in the presence of a reducing agent, and optimization of reaction conditions in order to achieve stability of synthesized NPs, further subjected to characterization by using biophysical techniques such as UV-visible spectroscopy, TEM, scanning electron microscopy (SEM), and FTIR. Flower-induced NPs may possess specialized properties, including antimicrobial as well as antioxidant activities [33,34,35] that may be useful in mediating abiotic stress tolerance, and therefore considered significant for agricultural practices. Until now, flower-mediated green synthesis of NPs has mostly been restricted to AuNPs and AgNPs [62]. Other potential plant sources for green synthesis as well as metallic NPs that are being synthesized must be explored.

### 2.6. Fruit

Fruit and peel extracts have been widely used as reducing and stabilizing agents in the synthesis of NPs over the last decade. Both dried and powdered fruit extracts have also been used for the synthesis of NPs. Sujitha and Kannan [63] performed TEM analysis and reported the synthesis of Au-nanoprisms, nanotriangles and nanospheres from citrus fruit extracts of *Citrus limon*, *C. reticulate* and *C. sinensis*. In the study, the citrus fruit extracts obtained from different citrus species were served to reduce the tetrachloroaurate (AuCl_4_) ions for the synthesis of AuNPs. Ghaffari-Moghadda and Hadi-Dabanlou [64] reported the production of Ag-nanospheres by using *Crataegus douglasii* fruit extract, which showed antibacterial activity on both Gram-positive and Gram-negative bacteria. Ramesh et al. [65] reported that AgNPs could be prepared using *Emblica officinalis* fruit extract, and FTIR studies showed that phytochemical constituents such as alkaloids, phenolic compounds, amino acids, carbohydrates and tannins act as reducing agents in the preparation of AgNPs. Several NPs synthesized by green synthesis using fruit extract possess anti-inflammatory properties, as reported by AgNPs synthesized from *Sambucus nigra* [66] and *Ficus carica* [38] fruit extracts. In addition, the anti-cancer activity of AgNPs synthesized by *Cleome viscose* fruit extract as reported by Lakshmann et al. [36], and the anti-viral activity of CuONPs synthesized by *S. alternifolium* fruit extract by Yugandhar et al. [37] were also reported.

### 2.7. Seeds

Seeds provide a zero energy-based, non-toxic, environmentally friendly, and cost-effective approach for the synthesis of NPs. Seed extracts of a large number of plant species have been used for the green synthesis of NPs, as they provide a medium for the reduction and stabilization of NPs that are being synthesized. Seeds have a high source of phytochemicals such as carbohydrates, phenolic compounds and reducing sugars that play a crucial role in the reduction and formation of NPs [39,67]. Variation in shape and size of NPs being synthesized can be controlled by altering the concentration of seed extract. Dhand et al. [39] reported the formation of spherical/ellipsoidal AgNPs after TEM analysis by making the use of *Coffea italica* seed extract. Seed exudates, seed powder or seed endosperm of *Sinapis arvensis* are also used for the purpose of AgNPs synthesis [67]. Moreover, ZnONPs and Fe_2_O_3_NPs synthesized using seed extracts of *Peganum harmala* and *Punica granatum* have an excellent adsorptive potential for chromium ions and photocatalytic properties, respectively [40] (Table 1).

## 3. Green Synthesized Nanoparticles-Mediated Reprogramming of Plant Traits

Studies on phytogenic NPs solely emphasized their role in the preliminary stages of the plant life cycle, i.e., seed germination, and reported improving the growth and development of NPs treated plants (Table 2; Figure 3). Titanium dioxide NPs synthesized from seed extract of *Cuminum cyminum* significantly increased the germination indices of *Vigna italica* at 50 µg mL^−1^ [68]. Similarly, AuNPs (2000 mg L^−1^) synthesized from leaf extracts of *Tiliacora triandra* positively modulated the germination rate in *Oryza sativa* [69]. In another experiment, AuNPs (5.4 µg mL^−1^) synthesized from *Allium cepa* extract were found to be beneficial in increasing the seedling emergence and average yield of the same plant [70]. Additionally, cerium oxide NPs (Ce_2_O_3_NPs) synthesized using leaf extracts of *Elaeagnus angustifolia* improved the growth and metabolic rate in *Solanum lycopersicum* at 20 and 100 mg L^−1^ concentrations [71]. Elakkiya et al. [72] studied the effect of CuONPs synthesized using *Sesbania italica* leaf extract on *Brassica nigra*, and observed an increase in the plant growth at 25 and 30 mg 100 mL^−1^ of CuONPs treatment, thus indicating their role in sustaining the crop productivity and yield. Rani et al. [73] revealed that ZnONPs and MgONPs synthesized from *Aloe barbadensis* leaf extract increased the germination rate, root-shoot length, and plant biomass in both *Vigna radiata* and *Cajanus cajan* at 5 mg concentration. Gold NPs synthesized from *Terminalia arjuna* fruit extract enhanced the node elongation, the number of leaves and lateral roots, and the total fresh weight of rhizome in the endangered medicinal plant *Gloriosa superba* at 1000 µM [74]. In *Hordeum vulgare,* treatment with Fe_2_O_3_NPs synthesized from *Cornus mas* fruit extract at 10–100 mg L^−1^ resulted in increased root-shoot biomass, and hence plant growth [75]. In addition, there is much evidence that gives systematic and constructive overviews on nutrient sequestration, photosynthetic pigments assimilation, bio-molecular metabolism, and antioxidant activities in response to NPs’ supplementation in plants. Zhang et al. [76] reported that the foliar application of AgNPs (50 mg L^−1^) on *Cucumis sativus* green synthesized from the *C. sativus* leaves and *O. sativa* husk extracts increased the protein and manganese (Mn) contents, respectively, indicating the positive effect of these NPs on nutrient retention and acquisition. In the same experiment, the chlorophyll content and photosynthetic rate were also enhanced. Similarly, sulfur NPs (SNPs) synthesized from *Punica granatum* peel extract improved the nutrient content of *S. lycopersicum* fruits at 200 ppm [77]. In *Zea mays,* the *Alpinia italica* rhizome extract-synthesized AuNPs positively influenced sugar, protein and sucrose levels at 10 ppm concentration [78]. The treatment with SNPs synthesized from *Cinnamomum zeylanicum* bark extracts improved the osmolyte (proline, glycine betaine, soluble sugars), and phytochemicals (anthocyanin, tannin, total phenol, flavonoid) contents in *Lactuca sativa* at 1 mg mL^−1^ concentration. Furthermore, the reduced levels of the stress markers such as malondialdehyde (MDA) and hydrogen peroxide (H_2_O_2_) suggested the potential application of NPs treated plants in sustaining the growth and photosynthetic parameters, even under stressful environmental conditions [79]. Similar results were also reported in *Z. mays* upon exposure to biogenic ZnONPs (50 mg L^−1^) synthesized using plant extract of *Lemna minor* [80]. In *Phaseolus vulgaris*, the application of biogenic AgNPs produced by leaf extracts of *Thuja occidentalis* enhanced the leaf number, leaf area index (LAI), chlorophyll content, nitrate reductase (NR) activity, and pod yield at 25–50 mg kg^−1^ [81]. In another experiment, AgNPs (100 mM) synthesized from *E. globules* leaves improved the overall growth of *Z. mays*, *A. cepa*, and *Trigonella foenum-graecum* by increasing the seed germination, antioxidant enzymatic activities (catalase, CAT; peroxidase, POX; ascorbate peroxidase, APX) and non-enzymatic (ascorbate, AsA; glutathione, GSH) contents [82]. In addition, Tripathi et al. [83] studied the impact of AgNPs (1 mg L^−1^) biosynthesized from *Withania coagulans* on the plant growth and withanolides production of the same plant. This study implies that low concentrations of synthesized AgNPs considerably improved growth and up-regulated the expressions of withanolides biosynthetic genes, and antioxidant genes superoxide dismutase (*SOD*; *CuZnSODc, MnSODc*), *CAT*, glutathione reductase (*GRc*), and *APX*. In *Brassica rapa*, the application of AgNPs synthesized from *V. negundo* plant extract induced the over-expression of anthocyanin regulatory genes such as anthocyanins pigment-1 (*PAP1*), anthocyanidin synthase (*ANS*), phenylalanine ammonia lyase (*PAL*), and hence resulted in anthocyanin accumulation in *B. oleracea* [84]. Furthermore, glucosinolates (GSLs) regulatory (*BrMYB28, BrMYB29, BrMYB34, BrMYB51*) and biosynthetic (sulfotransferase-5C, *ST5C*; C-S lyase sulfurylase-1, *SUR1*) gene expressions were also up-regulated on AgNPs application. Abbasifar et al. [85] found that combined applications of ZnNPs and CuNPs (1000, 2000 and 4000 ppm) synthesized from *O. basilicum* extract increase the photosynthetic pigments of the same plant, suggesting their participation in pigment biosynthesis. Additionally, the synergistic effect of ZnNPs + CuNPs showed the highest 2,2-diphenylpicrylhydrazyl (DPPH) radical scavenging activity at 4000 ppm and 2000 ppm, respectively. In addition, Fe_2_O_3_NPs biosynthesized using the flower extract of *Hydrangea paniculata* enhanced the seed vigor index (SVI), root morph-metric traits and enzymatic antioxidants activities (POX and CAT) at 1000 mg L^−1^ in *Linum usitatissimum* [86]. Cuppor oxide NPs prepared from the whole plant extract of *Adiantum lunulatum* enhanced the total phenolic and flavonoid contents at 0.025 mg mL^−1^ in *Lens culinaris* [87]. In addition, defensive enzymatic activities such as polyphenol oxidase (PPO), phenylalanine ammonia lyase (PAL), POX, APX, SOD, and CAT were increased at 0.05 mg mL^−1^ in *L. culinaris*. This study deliberately gave the indicative role of NPs in triggering the hypersensitive reaction associated with the cross-linking and lignification of the cell wall, activation of phenylpropanoid pathway and up-regulated antioxidants activities in the treated plants.

Green synthesized NPs can significantly alter the enzymatic and non-enzymatic antioxidant defense system and thus impact ROS homeostasis. Such amendments in cell functioning can help plants to adapt for the prevention of oxidative damage, especially to cellular membranes. The application of SNPs (0.01–1 mg L^−1^), synthesized from *C. zeylanicum* bark extract, to *L. sativa* reduced the levels of stress markers such as MDA and H_2_O_2_ [79]. Similar results were also reported in *Z. mays* upon exposure to phytogenic ZnONPs (50 mg L^−1^) synthesized using plant extract of *L. minor* [80]. In another experiment, AgNPs (0.05 mg L^−1^) synthesized from *E. globulus* improves the overall oxidative defense of *Z. mays*, *A. cepa* and *T. foenum-graecum* via increasing the antioxidant enzymatic activities (CAT, POX and APX) and non-enzymatic antioxidant (AsA and GSH) contents [82]. Furthermore, there was a reduction in MDA content, which highlighted the efficiency of green synthesized AgNPs in membrane stabilization, associated with decreased lipid peroxidation and oxidative stress-induced adversities. In addition, defensive enzymatic activities such as PPO, PAL, SOD, POX, APX, and CAT were increased at 0.05 mg mL^−1^ in *L. sativa* upon the exogenous application of green synthesized CuONPs [87]. Peroxidase and CAT activities were also reported to be up-regulated in *L. usitatissimum* treated with Fe_2_O_3_NP (1000 mg L^−1^), biosynthesized using the flower extract of *H. paniculata* [86]. Abbasifar et al. [85] reported that the synergistic effect of biosynthesized ZnNPs + CuNPs (1000, 2000 and 4000 ppm) showed improved antioxidant capacity in *O. basilicum.*

In addition to the enhancement of the antioxidant defense system, phytogenic NPs application can also alter the antioxidant gene expression levels in response to excess ROS production. For instance, Tripathi et al. [83] explored the impact of phytogenic AgNPs on gene expression levels in *W. coagulans* in response to the applied AgNPs (1 mg L^−1^) synthesized from *W. coagulans* leaves, which up-regulated the mRNA expression level of enzymatic antioxidants such as *SOD* (*CuZnSODc, MnSODc*), *CAT*, glutathione reductase (*GRc*), and *APX* genes in *W. coagulans.*

## 4. Role of Green Synthesized Nanoparticles in Alleviating Abiotic Stresses

In addition to playing a critical role in reprogramming the growth and development of plants, green synthesized NPs are also widely known to be implicated in the regulation of abiotic stress management [88,89]. Table 3 and Figure 4 summarize the main roles of green synthesized NPs under different abiotic stresses.

### 4.1. Salinity Stress

Salinity stress affects the primary metabolic mechanisms of plants through osmotic and ionic stresses [97]. The excessive accumulation of sodium (Na^+^) and chloride (Cl^−^) ions disrupts the ionic equilibrium, cellular metabolism, and membrane dysfunction, and hence delimits plant growth and development [98]. To mitigate these antagonistic responses, maintaining the ionic homeostasis and osmotic potential serve as the crucial parameter for inducing salinity tolerance responses in stressed plants. Extensive research has elucidated the role of green synthesized NPs in alleviating the salinity stress-induced adversities in plants (Table 3). Habibi and Aleyasin [91] reported that SeNPs (100 ppm) synthesized from *H. vulgare* leaves were effective in ameliorating the detrimental effects of salinity stress by increasing root and shoot traits (length, fresh weight, and dry weight), flavonoids, phenolic, and photosynthetic pigment contents as well as reducing the stress markers (MDA and H_2_O_2_).

The amelioration of salinity stress by improved growth and physiological functioning upon the application of phytogenic NPs has been supported by numerous other studies. For instance, Zafar et al. [99] revealed that exogenously applied zinc NPs (ZnNPs) synthesized from *Sorghum bicolor* leaf extract improved growth and photosynthetic traits in salinity stressed *Abelmoschus esculentus* at 0.3% concentration through up-regulating the antioxidant defense system. Similarly, under salinity stress, foliar application of ZnONPs (10 mg L^−1^) synthesized from leaflet extract of *Phoenix dactylifera* improved growth characteristics, plant growth rate, biomass production, and elevated the activities of ROS-detoxifying enzymes (SOD and CAT) in *V. unguiculata* and *A. esculentus* [100,101]. Similarly, TiO_2_NPs (40 mg L^−1^) synthesized from leaf extract of *Buddleja asiatica* ameliorated salinity stress-induced alterations in *T. aestivum* by improving various morpho-physiological and biochemical attributes [102]. In accordance with this, Ag-Au alloy NPs (100 ppm) synthesized from rhizome extract of *Mentha piperita* positively altered the growth and biochemical traits in *M. piperita* under salinity stress [103]. Calcium oxide NPs (CaONPs) synthesized using *Juglans regia* (shell) also enhanced the germination rate of *P. vulgaris* under salinity stress at 50 mg kg^−1^ [104], thus indicating the ability of green synthesized NPs in mediating germination efficiency under imposed stress condition.

Plant extract-based synthesis of NPs has also proven to be an effective supplement in inducing stress-responsive signaling. Calcium oxide NPs (1.5 ppm) synthesized from *P. granatum* fruit extract were found to be significant in alleviating the salinity stress-induced adversities in *Triticale* callus [105]. In this study, the confocal laser scanning microscopy revealed the accumulation of calcium ions (Ca^2+^) in *Triticale* cultivars (*Tathak, Umran Hanum, Alper Bey*), indicating that CaONPs might act as stress signaling transducers for Ca^2+^-mediated plant stress responses under the salinity stress conditions. Interestingly, the NPs have also been reported to regulate nutrient homeostasis and are known to provide protection from ionic toxicity in plants during salinity stress [93]. For instance, foliar application of biosynthesized AuNPs imposed a significant impact on shoot and root ionic contents, as well as improved nitrogen (N) metabolic activity, and enzymatic antioxidant activities (SOD, APX, GPX, and GR) as well as non-enzymatic antioxidant contents (AsA and GSH), with reduction in ROS generation and lipid peroxidation under salinity stress [93] (Table 3). Ragab and Saad-Allah [106] found that SNPs (50, 100 and 200 µM) derived from *O. basilicum* leaves increased the nutrient content including N, phosphorous (P) and potassium (K), along with an improved ionic ratio of K^+^: Na^+^, and also promoted the acquisition of cysteine, free amino acids and total soluble proteins in *T. aestivum* under salinity stress. Furthermore, Yasmin et al. [92] evaluated the influence of ZnONPs (17 mg L^−1^) synthesized from *Carica papaya* extract on the antioxidant metabolism in *Carthamus tinctorius* under salinity stress, and revealed that ZnONPs up-regulated the activity of antioxidant enzymes and proline content, while they hampered the ROS production (H_2_O_2_ ,O_2_^−^, superoxide radical; and MDA) against imposed salinity stress.

### 4.2. Drought Stress

Drought is perhaps one of the most significant abiotic stresses that are responsible for reducing crop productivity and quality. It occurs due to changes in temperature dynamics, light intensity, and low rainfall resulting in water-deficit conditions [107]. The induction of drought stress triggers a wide range of plant responses that include alteration in growth traits, biochemical responses including enzymatic antioxidant activities, protein and metabolite contents [108]. The application of phytogenic NPs is a promising strategy to ameliorate the detrimental effects of drought stress in plants (Table 3). For instance, *Chaetomorpha antennina* was used in the synthesis of iron NPs (FeNPs) which were bare and some were coated with citrate compounds [18]. These NPs were reported to be efficient in enhancing drought stress tolerance in *Setaria italica* through the positive modulation of enzymatic antioxidant activities such as CAT, SOD and POX, and osmolytes concentrations. Furthermore, FeNPs also improved the photosynthetic efficiency, growth traits, and diverse biochemical responses in *S. italic* at 50–120 mg L^−1^ concentration (Table 3). Furthermore, SeNPs (30 mg L^−1^) prepared using *A. sativum* bud extract were applied to *T. aestivum* in response to drought stress. These SeNPs significantly improved the growth parameters (plant height, biomass accumulation, leaf area, number, and length) while reducing ionic leakage and lipid peroxidation, thus aiding in ameliorating the drought stress-induced cellular toxicity [109].

### 4.3. Heat Stress

In the past few years, heat stress has emerged as one of the potent abiotic stresses associated with climate change and it has a detrimental impact on crop production around the world [110]. Heat stress causes leaf burning, abscission, fruit damage and senescence as well as decreased plant growth (shoot and root) and productivity [111]. However, green synthesized NPs can be effective in ameliorating the detrimental effects imposed by heat stress on plants. There is very little evidence that depicts the NPs mediated acclimatized responses in plants under high temperature stress. Iqbal et al. [19] used AgNPs synthesized from leaf extract of *Moringa oleifera*, and reported that the application of AgNPs at 50 and 75 mg L^−1^ concentrations played a pivotal role in mitigating the adverse impacts of heat stress in *T. aestivum* by lowering the contents of MDA and H_2_O_2_, along with improved antioxidant defense system (Table 3).

### 4.4. Heavy Metal Stress

Rapid globalization significantly contaminates the environment through the emission of higher levels of toxic metals. Once deposited in soils, heavy metals exert a negative impact on soil dynamics [107] and microbial structural organization [112], resulting in decreased soil fertility and crop efficiency [113]. Heavy metal stress negatively affects the water potential, photosynthetic efficiency, and growth attributes, and often leads to crop failure [113]. However, various studies have elucidated the potentiality of phytogenic NPs in inducing tolerance mechanisms by regulating the overall plant growth and physiological functioning in response to heavy metal stress (Table 3). For instance, Venkatachalam et al. [16] examined the effects of ZnONPs (25mg L^−1^) on *Leucaena leucocephala*, synthesized using *Ulva lactuca*, in response to imposed heavy metal stress including lead (Pb; 100 mg L^−1^) and cadmium (Cd; 50 mg L^−1^). These green synthesized ZnONPs significantly increased the plant growth and photosynthetic pigments in heavy metal stressed *L. leucocephala*. Similarly, the treatment with Fe_3_O_4_NPs (0.5 mg g^−1^) synthesized from husk extract of *Cocos nucifera* in Cd-stressed *O. sativa* enhanced the plant biomass, quantum efficiency of photosystem II (PSII) and chlorophyll content and improved the crop productivity [95] (Table 3). Another analysis revealed that Fe_3_O_4_NPs (0.5 g) synthesized from the bark extract of *Hevea barsiliensis* increased plant biomass, photosynthetic pigments, and maintained nutrient homeostasis, along with reducing the accumulation of Cd^2+^ ions, and stress-induced oxidative damages in *O. sativa* [96] (Table 3). Foliar application of TiO_2_NPs (100 mg L^−1^) synthesized from leaf extract of *T**rianthema portulacastrum* and *Chenopodium quinoa* prominently inhibited the assimilation of Cd^2+^ ions in the plant system and was found to be beneficial in improving the plant height, spikes’ length, chlorophyll content, and grain yield of *T. aestivum* [114]. The application of SNPs (100 µM) synthesized from *O. basilicum* leaf extract provides tolerance against Mn (100 mM)-induced stress responses and increases the contents of crude protein, total amino acid and cysteine, with decreased ROS-mediated lipid peroxidation in *Helianthus annus* [115]. Similarly, the exogenous application of TiO_2_NPs (0.1%) synthesized using leaf extract of *Musa paradisica* in arsenic (As) stressed *V. radiata* significantly decreased the accumulation of As^3+^ ions, and enhanced the protein content, along with increasing enzymatic antioxidant activities (SOD, CAT and APX) which promoted ROS detoxification [94]. Furthermore, ZnONPs (25 mg L^−1^) play a significant role in the activation of the ROS-scavenging system (SOD, CAT, and POX), and hindered Pb-elicited physio-biochemical changes [16].

## 5. Nanoparticles Modulate Phytohormones under Abiotic Stress

Phytohormones are crucial chemical messengers that facilitate numerous cellular functions by regulating several metabolic pathways, and therefore exert a potential influence on plant growth and development [116]. The key findings emphasized the importance of green synthesized NPs in regulating phytohormone content under abiotic stress conditions, and have been summarized below.

Crosstalk between different phytohormones and plant synthesized NPs in response to abiotic stress conditions has been a major challenge for plant researchers [117]. Several studies have revealed that the signaling pathways of NPs and plant growth regulators (PGRs) including phytohormones are interconnected during abiotic stress-induced responses, which exclusively influenced plant growth and physiological attributes. The exogenously applied CuNPs and AgNPs (250, 750 and 1000 ppm), biosynthesized from *Justicia spicigera* leaf extract, promoted root development in *Annona muricata* by modulating endogenous concentrations of indole-3-acetic acid (IAA) and gibberellic acid (GA_3_) [118]. Iron NPs (20, 40, 80 and 160 mg L^−1^) synthesized from *A. cepa* extract significantly increased the accumulation of photosynthetic pigments, and non-enzymatic antioxidant compounds, along with stimulating the biosynthesis of jasmonic acid (JA) by significantly increasing the assimilation of 12-oxophytodienic acid (OPDA; JA-precursor) in diploid and triploid varieties of *Citrullus lanatus* seedlings [119]. Wahid et al. [7] reported that foliar application of AgNPs (300 ppm) synthesized from leaf extract of *T. aestivum*, triggered the antioxidant defense system, and modulated proline metabolism and N assimilation in *T. aestivum* under salinity stress. These green synthesized AgNPs resulted in decreased abscisic acid (ABA) concentration while exerting a positive influence on chlorophyll content and stomatal dynamics, and lowered the accumulation of stress indicators such as H_2_O_2_ and thiobarbituric acid reactive substances (TBARS), which eventually ameliorated the detrimental impacts of salinity stress in *T. aestivum* [7]. Abou-Zeid and Ismail [90] reported that AgNPs (1 mg L^−1^) synthesized from *Capparis spinosa* extract increased salinity stress tolerance in *T. aestivum* by preventing oxidative stress-induced cellular damages, which aid in improving plant growth traits and photosynthetic efficiency. Moreover, AgNPs significantly modulated phytohormone homeostasis by enhancing indole-3-butyric acid (IBA), 1-naphthalene acetic acid (NAA) and 6-benzylaminopurine (BAP) contents, while decreasing ABA content. Mustafa et al. [17] studied the impact of exogenously applied TiO_2_NPs (40 ppm) in *T. aestivum,* which stimulated the interactions between TiO_2_NPs and phytohormone (IAA and GA), resulting in increased photosynthesis, amino acid and carbohydrate metabolism, as well as enhancing the antioxidant enzymatic activities (SOD, POX and CAT), thus subsequently promoted the drought stress resilience traits in *T. aestivum*. In general, the observed regulation of endogenous phytohormone contents in response to the application of green synthesized NPs is consistent under both stressed and stress-free or optimal conditions. Overall, more studies that highlights the interplay between NPs and phytohormones are needed to gain a better understanding of their mechanistic actions in response to various abiotic stresses, which may shed light on their effective use in agricultural sustainability.

## 6. Applications and Limitations of Green Synthesized Nanoparticles

Plant-based synthesis of NPs is a green strategy that fills the gap between nanotechnology and plant sciences, and is gaining immense popularity because of their economical, non-toxic, and eco-friendly nature as compared to chemically synthesized NPs. Phyto-nanotechnology has a promising future in the production of NPs from plant extracts such as root, stem, leaves, flowers, fruits and seeds.

Green synthesized nanomaterials play a determinant role in the development of long-term technologies for humanity as well as for the environment due to their potential applications in the pharmaceutical sectors, electronics, environment remediation approaches, and biomedical fields (Figure 5). For example, AuNPs have been reported to be used in medical sciences, primarily due to their high affinity for a wide range of biomolecules [120]. In addition, AuNPs have several biomedical applications, including enzyme modulation and antibacterial activity [121]. In addition to AuNPs, AgNPs also have been extensively studied and reported to act as both antibacterial and antifungal agents in agronomic industries and pharmaceutical sectors, and have been utilized in food packaging products [122]. Silver NPs are also exploited as a carrier of bioactive compounds in anticancer therapies and are beneficial for the diagnosis of diseases, particularly cancer [31]. Velmurugan et al. [24] reported the application of AgNPs as a strong antibacterial agent against both Gram-negative and Gram-positive bacteria. Nickel oxides NPs have been recognized as a photocatalytic agent, and they possess a strong adsorptive capacity for dye and pollutants [26,29]. Iron NPs were proved to be an efficient agent for wastewater treatment, as they aid in the removal of phosphates and nitrates, and thus increase the chemical oxygen demand (COD) in water bodies [22,123].

Palladium NPs are one of the most valuable and rare high-density metals widely used as a catalyst and biosensor for medical diagnostic purposes [124]. It has the potential to effectively catalyze a wide range of chemical reactions and enhances the yield of desired products. Notably, ZnONPs have revealed their widespread use for agronomic interest by exhibiting anti-phytopathogenic activity against both fungi and bacteria [125]. Moreover, ZnONPs showed applications in diverse areas of medicine and drug delivery systems [28], along with excellent photocatalytic activity and absorption capacity for heavy metal (Cr), thus showing their ability to detoxifying the variety of organic and inorganic pollutants present in the environment [40]. Moreover, MgONPs were found to be effective in eradicating organic dyes such as methylene blue (MB), probably due to their active surface area, strong reactivity and affinity for several chemical compounds [126]. Metal oxide NPs including CeO_2_NPs act as a strong chelating agent and were found to be effective in carrying out several chemical reactions [33]. Earth metals such as Nd_2_O_3_NPs have been reported to possess anti-inflammatory, anti-diabetic activity as well as antioxidant properties [30].

However, green fabricated NPs faces multiple constraints and limitations, which hinders the novel interface between nanotechnology and agro-environment sustainability. Green synthesis of NPs experiences setbacks regarding the selection of plant materials, synthesis conditions, product quality, control and their applications [127]. For instance, some plant materials are available in abundance in the endemic regions only, which makes the plant extract collection a tedious procedure and hinders the large-scale global production of bio-compatible NPs. In addition, the excessive energy consumption, long reaction period and use of industrial chemical compounds as oxidizing and/or reducing agents during NPs synthesis make it a challenging task to achieve. After the synthesis of NPs, characterization of NPs regarding shape and size serves as the crucial parameters for determining the quality of synthesized NPs. Since different plant extracts are used in the synthesis of NPs, there is a lack of understanding of NPs assessment due to genetic variability within or between the plant species, and thus there is a necessitated requirement of high-throughput instrumentations for the purposes of NPs purification and characterization. Moreover, the conversion rate and yield of NPs during synthesis is comparatively low compared to chemically synthesized NPs, and thus subsequently lowers the economic benefits. Due to the extremely positive outcomes of green synthesized NPs, it is vital and necessary to take steps to analyze the limitations/challenges imposed by several factors during NPs synthesis and to rectify them. For instance, the use of ideal raw plant material or substitute material instead of indigenous or seasonal plants, reduced usage of technologies that consume high-energies, product optimization, and storage of nanoscale products for long period, and practical difficulties in the synthesis of NPs as well as their applications should be avoided with innovative scientific notions.

## 7. Conclusions and Future Perspectives

In conclusion, it is clear that over the last decade, there has been an intensifying demand for green chemistry and the use of green methods for the synthesis of plant-based NPs, which has led to a goal to promote eco-friendly techniques. Several studies have been conducted on plant extract-mediated synthesis of NPs and their promising applications in different fields. This is probably due to the unique characteristics of NPs as an inert molecule, ease of accessibility, and their environmentally benign nature that imposes low economic as well as ecological constraints on the surrounding environment. In this review, we have addressed the potential use of plant extracts as the source for NPs synthesis, as well as significant applications of NPs in environment remediation strategies such as wastewater treatment, pharmaceutical sectors, and biomedical fields. Furthermore, this review emphasizes the efficient role of green synthesized NPs in reprogramming the plant traits, including seedling germination, growth parameters, photosynthetic efficiency, mineral uptake, and yield attributes under stressed and non-stressed conditions. In addition, NPs up-regulate the antioxidant defense system and the accumulation of compatible solutes, which decreases oxidative stress-induced ROS accumulation, thereby aiding in acquiring tolerance traits against different abiotic stresses. Recent studies have also focused on the role of green synthesized NPs in modulating phytohormones in response to abiotic stress. Green synthesized NPs serve as an intriguing and evolving aspect of nanotechnology that has a significant impact on the environment in terms of sustainability and future advancement, and it is expected that there are surplus applications of NPs that will be exercised in the upcoming years. However, a few challenges or limitations associated with the green synthesis approaches should be addressed by the researchers. Further research is needed to explore the precise molecular mechanisms, which elicit the intra-cellular plant signaling responses with NPs application under stress or optimal conditions.

## Figures and Tables

**Figure 1 ijms-23-04452-f001:**
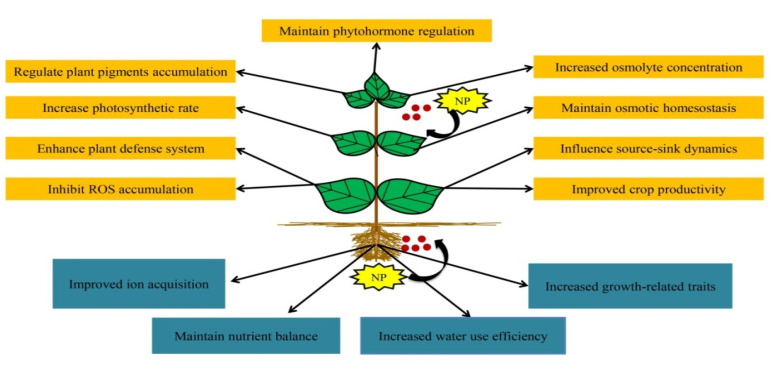
A summary of physiological and biochemical responses in plants on exogenous application of green synthesized nanoparticles by foliar application or direct administration in the soil.

**Figure 2 ijms-23-04452-f002:**
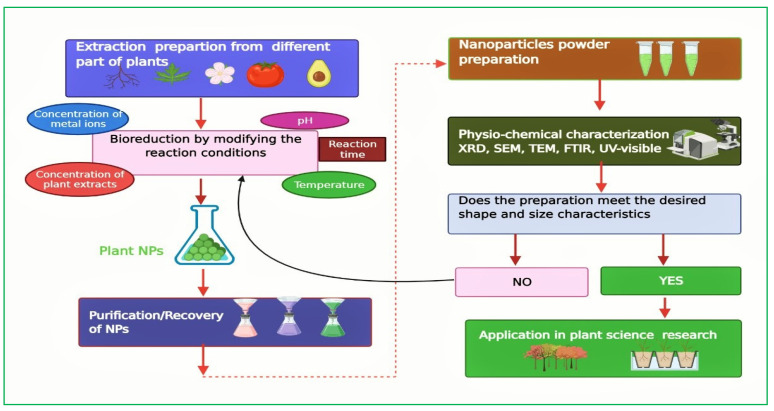
Schematic diagram showing the preparation of green synthesized nanoparticles from different parts of the plant. Plant extract preparation from different plant parts such as roots, leaves, flower, fruits and seeds are used in the synthesis of nanoparticles (NPs). The bio-reduction mediated synthesis of NPs is controlled by several factors including concentration of plant extracts and metal ions, pH of the solution, reaction time, and temperature at which reaction is carried out. Purifications and characterization of NPs play a determinant role in the synthesis of desired NPs, which could be beneficial in plant science and research-oriented disciplines. The reaction should be restarted from the bio-reduction process if the synthesized NPs do not meet the desired morphological characteristics. Black, red and dotted arrows show the steps involved in NPs synthesis.

**Figure 3 ijms-23-04452-f003:**
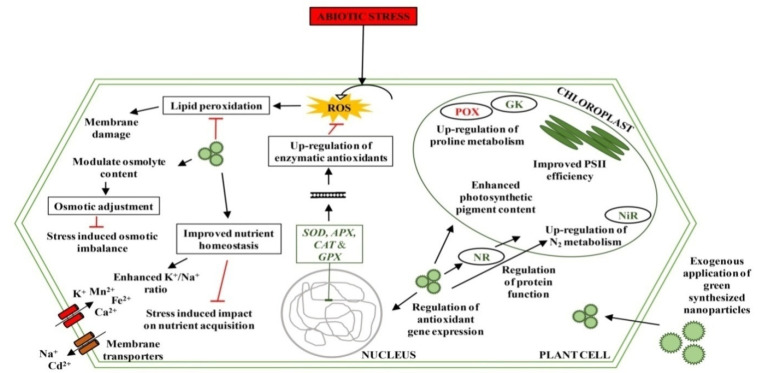
Summarization of cellular parameters regulated by the application of green synthesized nanoparticles to plants under different abiotic stresses. The presented NPs are a generalization of various green synthesized NPs administered in abiotic stressed plants, and reported to impart tolerance traits by regulating the various cellular/physiological aspects. Membrane transporters bring about nutrient uptake and efflux, the efficiency of which is hampered by various abiotic stress conditions but is restored and maintained by green synthesized NPs’ supplementation. Enzymatic antioxidants up-regulated by the action of green synthesized NPs aid in ameliorating abiotic stress-induced oxidative damages and maintain cellular homeostasis. Photosynthetic efficiency, nitrogen metabolism and osmolyte concentrations were enhanced upon NPs’ application, and thus play a crucial role in imparting abiotic stress tolerance. Genes and proteins represented in green are up-regulated, while those represented in red are down-regulated. Black arrow lines represent effects imposed by NPs and/or a further step in a natural series of cellular events. Red line with a flat head represents the repression effect. *APX*, ascorbate peroxidase gene; Ca^2+^, calcium ions; *CAT*, catalase gene; Cd^2+^, cadmium ions; Fe^2+^, ferrous ions; GK, glutamyl kinase; *GPX*, glutathione peroxidase gene; K^+^; potassium ions; Mn^2+^, manganese ions; Na^+^, sodium ions; NiR, nitrite reductase; NR, nitrate reductase; NPs, nanoparticles; POX, proline oxidase; ROS, reactive oxygen species; *SOD,* superoxide dismutase gene.

**Figure 4 ijms-23-04452-f004:**
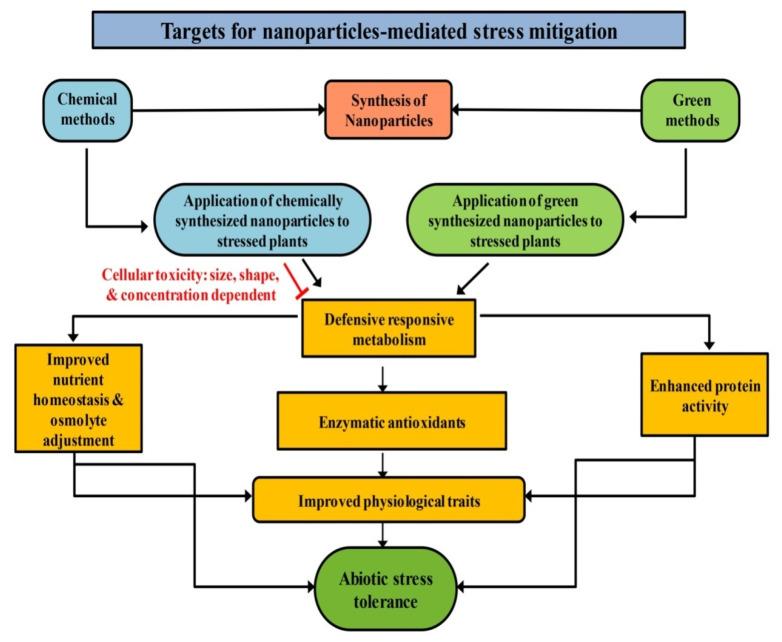
Potential targets for nanoparticles-mediated abiotic stress mitigation. The synthesis of nanoparticles (NPs) by green approaches is an environment-friendly method and can be efficiently used to trigger various abiotic stress-induced responses upon exogenous application to stressed plants. Chemically synthesized NPs show nanotoxicity and limit the ameliorative responses for abiotic stress alleviation (indicated by the red inhibitory arrow). Strategies focused on eliciting plant defense responses by triggering osmotic adjustment, antioxidant machinery for amelioration of cellular damage by ROS, and improving protein activity, can be efficient in conferring resistance to various abiotic stresses. Green synthesized NPs prove to be a suitable candidate for stress mitigation as they help in altering the gene expression of enzymatic antioxidants (SOD, CAT, and GR), and enhance osmotic content and nutrient homeostasis. Furthermore, green synthesized NPs favor the enhancement in protein activity, particularly of enzymes involved in N and proline metabolism which help in accomplishing the abiotic stress tolerance in crops. CAT, catalase; GR, glutathione reductase; N, nitrogen; ROS, reactive oxygen species; SOD, superoxide dismutase.

**Figure 5 ijms-23-04452-f005:**
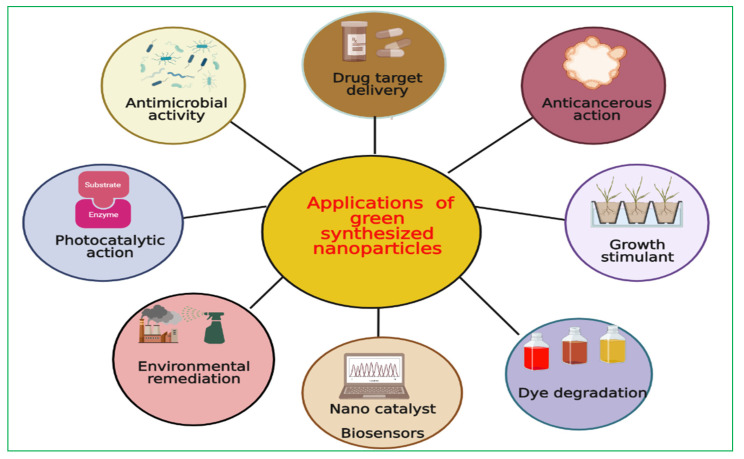
Applications of green synthesized nanoparticles in different fields. Black straight line represents the different applications of plant-synthesized nanoparticles in various fields of plant science and research-oriented disciplines.

**Table 1 ijms-23-04452-t001:** Green synthesis of various nanoparticles using different plant parts and their applications.

Plant Extract		Nanoparticles Characteristics and Applications				Ref.
Type	Scientific Name	Type	Size	Shape	Applications	
Root	*Zingiber officinale*	AgNPs	15 nm	Spherical	Antibacterial activity	[24]
Root	*Withania somnifera*	TiO_2_NPs	50–90 nm	Aggregates of spherical and square	Antibiofilm activity	[25]
Root	*Rheum turkestanicum*	NiONPs	12–15 nm	Hexagonal (rhombohedral)	Photocatalytic activity	[26]
Stem	*Leucas lavandulifolia*	SeNPs	56–75 nm	Spherical	Antibacterial Activity	[27]
Stem	*Mussaenda frondos*	ZnONPs	5–20 nm	Hexagonal (wurtzite)	Antioxidant activity and drug-target delivery	[28]
Leaf	*Leucas lavandulifolia*	SeNPs	56–75 nm	Spherical	Antibacterial Activity	[27]
Leaf	*Ocimum sanctum*	NiONPs	13–36 nm	Spherical to polyhedral	Pollutant adsorbent	[29]
Leaf	*Eucalyptus globulus*	Nd_2_O_3_NPs	50.37 nm	Smooth-surfaced particles with irregular particle shapes	Anti-inflammatory and antioxidant activity	[30]
Bud	*Tussilago farfara*	AgNPs AuNPs	13.57 ± 3.26 18.20 ± 4.11	Spherical	Anticancer agents	[31]
Bud	*Polianthus tuberosa*	AgNPs	50 ± 2 nm	Spherical (oval)	Larvicidal activity	[32]
Flower	*Hibiscus sabdariffa*	CeO_2_NPs	3.9 nm	Spherical	Chelating agent	[33]
Flower	*Callistemon viminalis*	Cr_2_O_3_NPs	92.2 nm	Cubic-like platelet	Oxidizing and/or reducing agent	[34]
Flower	*Rosmarinus officinalis*	MgONPs	8.8 nm	Round	Antibacterial Activity	[35]
Fruit	*Cleome viscose*	AgNPs	20–50 nm	Spherical	Antibacterial and anticancer activity	[36]
Fruit	*Syzygium alternifolium*	CuONPs	2–69 nm	Spherical	Antiviral activity	[37]
Fruit	*Ficus carica*	AgNPs	10–30 nm	Spherical	Antioxidant activity	[38]
Seeds	*Coffea arabica*	AgNPs	20–30 nm	Spherical and ellipsoidal	Antibacterial activity	[39]
Seeds	*Peganum harmala*	ZnONPs	40 nm	Non-uniform	Chromium (VI) adsorption	[40]
Seeds	*Punica granatum*	Fe_2_O_3_NPs	25–55 nm	Semi-spherical and agglomerated form	Photo-catalytic activity	[40]

AgNPs, silver nanoparticles; AuNPs, gold nanoparticles; CeO_2_NPs, cerium dioxide nanoparticles; Cr_2_O_3_NPs, chromium oxide nanoparticles; CuONPs, copper oxide nanoparticles; Fe_2_O_3_NPs, iron oxide nanoparticles; MgONPs, magnesium oxide nanoparticles; NiONPs, nickel oxide nanoparticles; Nd_2_O_3_NPs, neodymium oxide nanoparticles; SeNPs, selenium nanoparticles; TiO_2_NPs, titanium dioxide nanoparticles; ZnONPs, zinc oxide nanoparticles.

**Table 2 ijms-23-04452-t002:** Role of green synthesized nanoparticles in mediating reprogramming of plant traits.

Plant Extract	Nanoparticles	Size (nm)	Shape	Concentration	Studied Plant Species	Responses	Ref.
Seed of *Cuminum cyminum*	TiO_2_NPs	15.17	Spherical	50 µg mL^−1^	*Vigna radiata*	Significantly increased the growth attributes such as root and shoot length, germination percentage and rate, and mean daily germination.	[68]
Leaf of *Sesbania aculeata*	CuONPs	10.1	Spherical	25 and 30 mg 100 mL^−1^	*Brassica nigra*	Functioned as a strong antibacterial agent and bio-fertilizer for sustaining crop yield.	[72]
Fruit of *Cornus mas*	Fe_2_O_3_NPs	20–40	Spherical	10–100 mg L^−1^	*Hordeum vulgare*	Significantly improved root growth and shoot biomass.	[75]
Fruit of *Punica granatum*	SNPs	20	Spherical	200 ppm	*Solanum lycopersicum*	Increased plant growth and yield, and promoted accumulation of high-quality nutrients in fruits.	[77]
Root of *Alpinia galanga*	AuNPs	10–30	Spherical	5 and 10 mg L^−1^	*Zea mays*	Promoted emergence percentage and seedling vigor index.	[78]
Shoot apical meristem of *Withania coagulans*	AgNPs	14	Spherical	1 mg L^−1^	*Withania coagulans*	Improved root and shoot length, plant fresh weight, photosynthetic pigments, and anthocyanin contents.	[83]
Flower of *Hydrangea paniculata*	Fe_2_O_3_NPs	56	Spherical	1000 mg/L	*Linum usitatissimum*	Enhanced POX and CAT activities, and sustain plant growth.	[86]
Whole plant of *Adiantum lunulatum*	CuONPs	1.5–20	Quasi-spherical	0.025 and 0.05 mg m L^−1^	*Lens culinaris*	Enhanced the total phenolic and flavonoid contents and increased activities of PPO, PAL, POX, APX, SOD, and CAT enzymes.	[87]

AgNPs, silver nanoparticles; AuNPs, gold nanoparticles; APX, ascorbate peroxidase; CAT, catalase; CuONPs, copper oxide nanoparticles; Fe_2_O_3_NPs, iron oxide nanoparticles; PAL, phenylalanine ammonia lyase; PPO, polyphenol oxidase; POX, peroxidase; SNPs, sulfur nanoparticles; SOD, superoxide dismutase; TiO_2_NPs, titanium dioxide nanoparticles.

**Table 3 ijms-23-04452-t003:** Impact of green synthesized nanoparticles in amelioration of abiotic stress in plants.

Type of NPs	Plant extract used for NPs Synthesis	NPs Concentration	AbioticStress	Studied Plant Species	Plant Responses Under Abiotic Stresses	Ref.
AgNPs	*Capparis spinosa,* whole plant extract	1 mg L^−1^	Salinity (25 and 100 mM)	*Triticum aestivum*	Improved growth traits and photosynthetic responses, and increased IBA and BAP contents, along with decreased ABA concentration.	[90]
AgNPs	*Triticum aestivum;* leaf extract	300 ppm	Salinity (100 mM)	*Triticum aestivum*	Increased proline metabolism and nitrogen assimilation, and non-enzymatic antioxidant content such as AsA and GSH. Improved SOD, APX, GR, and GPX activities. Maintained the ionic homeostasis, and reduced ROS accumulation and lipid peroxidation.	[7]
SeNPs	*Hordeum vulgare;* leaf extract	100 ppm	Salinity (100 and 200 mM)	*Hordeum vulgare*	Increased contents of photosynthetic pigments, flavonoids and phenolic compounds, and hindered the accumulation of MDA and H_2_O_2_.	[91]
ZnONPs	*Carica papaya;* whole plant extract	17 mg L^−1^	Salinity (250 mM)	*Carthamus tinctorius*	Up-regulated the activity of antioxidant enzymes, and increased proline content while decreased the ROS production (H_2_O_2_, O_2_^.-^ radical and MDA). Improved yield attributes such as number of pods per plant.	[92]
AuNPs	*Triticum aestivum;* leaf extract	300 ppm	Salinity (100 mM)	*Triticum aestivum*	Increased nitrogen assimilation and antioxidant enzymatic activities and non-enzymatic compounds such as AsA and GSH, and maintained K^+^: Na^+^ ionic ratio in both root and shoot system.	[93]
FeNPs	*Chaetomorphaantennina*; whole algal extract	5, 10, 15, 20, 50, 90 and 120 mg L^−1^	Drought (10% PEG)	*Setaria* *italica*	Enhanced the accumulation of osmolytes, and activities of antioxidant enzymes such as CAT, SOD and POX. Improved photosynthetic efficiency, growth attributes, along with diverse biochemical responses.	[18]
AgNPs	*Moringa oleifera;* leaf extract	25, 50, 75, and 100 mg L^−1^	Heat (35–40 °C; 3 h/day)	*Triticum aestivum*	Lowered the contents of MDA and H_2_O_2_, and enhanced the antioxidant defense system.	[19]
TiO_2_NPs	*Musa paradisica;* leaf extract	0.1%	Heavy metal (arsenic; 10 µM)	*Vigna radiata*	Decreased accumulation of arsenic, while enhanced the protein content, and enzymatic antioxidant activities such as SOD, CAT and APX, and prevented ROS-induced adversities.	[94]
Fe_3_O_4_NPs	*Hevea* *barsiliensis*; barks extract	0.5 g	Heavy metal (cadmium; 15.0 mg kg^−1^)	*Oryza sativa*	Increased plant biomass, photosynthetic pigments, maintained nutrient homeostasis, and reduced cadmium-induced oxidative damages.	[95]
Fe_3_O_4_NPs	*Cocos nucifera;* husk extract	20 mg	Heavy metal (cadmium; 0.01% *w*/*w*)	*Oryza sativa*	Enhanced plant biomass, quantum efficiency of PSII, chlorophyll content, and increased crop productivity.	[96]

ABA, abscisic acid; AgNPs, silver nanoparticles; AsA, ascorbate; APX, ascorbate peroxidase; AuNPs, gold nanoparticles; BAP, 6-benzylaminopurine; CAT, catalase; FeNPs, iron nanoparticles; Fe_3_O_4_NPs, iron oxide nanoparticles; GSH, glutathione; GR, glutathione reductase; GPX, glutathione peroxidase; H_2_O_2_, hydrogen peroxide; IBA, indole-3-butyric acid; MDA, malondialdehyde; NPs, nanoparticles; O_2_^.-^, superoxide radical; K^+^: Na^+^, potassium and sodium ionic ratio; PEG, polyethylene glycol; PSII, photosystem II; POX, peroxidase; ROS, reactive oxygen species; SeNPs, selelenium nanoparticles; SOD, superoxide dismutase; TiO_2_NPs, titanium dioxide nanoparticles; ZnONPs, zinc oxide nanoparticles.

## Data Availability

Not applicable.

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
