# Peer review of "Bio-Synthesized Nanoparticles in Developing Plant Abiotic Stress Resilience: A New Boon for Sustainable Approach"

_ijms, 2022, doi:10.3390/ijms23084452_

Round 1

Reviewer 1 Report

Manuscript ID:  ijms-1663524

Title: Bio-synthesized nanoparticles in developing plant abiotic stress
resilience: A new boon for sustainable approach

General comments: The authors have reviewed the effects of bio-synthesized nanoparticles on abiotic stress tolerance in plants as a new sustainable approach. The review is meeting the scientific standard in terms of technical language, design and implication of the results. The English language of article seems better but some typo errors should be checked briefly during revision. I suggest minor revision for this article and will be happy to review its revised version. The following suggestions should be considered to improve the manuscript quality before its publication in International Journal of Molecular Sciences.

Specific comments:

1-     Please re-check the abbreviations which have been mentioned in the whole manuscript. Please elaborate them at least once at its first place in the abstract and other sections of manuscript.

2-     The figures are not of good quality. Some fonts are not readable. Please provide high quality figures.

3-     The introduction section is well written and in my opinion it sheds light on the problem in a concise manner. Overall, the objectives are briefly explained and are good to go with.

4- Please be checked the whole manuscript, the format should be improved such as there are some unit presentations, put the relevant unit with each value across the manuscript.

5- Please re-write the conclusion. The conclusion is recommended to be supported by the data shown in tables, put detail of any limitations of this study, describe implications of this study and provide recommendations for future perspectives.

Author Response

Reviewer#1      

General comments: The authors have reviewed the effects of bio-synthesized nanoparticles on abiotic stress tolerance in plants as a new sustainable approach. The review is meeting the scientific standard in terms of technical language, design and implication of the results. The English language of article seems better but some typo errors should be checked briefly during revision. I suggest minor revision for this article and will be happy to review its revised version. The following suggestions should be considered to improve the manuscript quality before its publication in International Journal of Molecular Sciences.

  • Author’s response- we thank the Reviewer for your appreciation

Specific comments:

  • Please re-check the abbreviations which have been mentioned in the whole manuscript. Please elaborate them at least once at its first place in the abstract and other sections of manuscript.
  • Author’s response- As per the reviewers’ comment, all the abbreviations have been elaborated in the entire manuscript.
  • The figures are not of good quality. Some fonts are not readable. Please provide high quality figures.
  • Author’s response- As per the reviewers’ comment, high-quality figures have been added.
  • The introduction section is well written and in my opinion it sheds light on the problem in a concise manner. Overall, the objectives are briefly explained and are good to go with.
  • Author’s response- we thank the Reviewer for your appreciation.
  • Please be checked the whole manuscript, the format should be improved such as there are some unit presentations, put the relevant unit with each value across the manuscript.
  • Author’s response- As per the reviewers’ comment, whole manuscript along with units has been checked and improved.
  • Please re-write the conclusion. The conclusion is recommended to be supported by the data shown in tables, put detail of any limitations of this study, describe implications of this study and provide recommendations for future perspectives.
  • Author’s response- As per the reviewers’ comment, the conclusion part has been re-written and section 6 has been added with implications and limitations of the study.

**************************************************************************

Reviewer 2 Report

The theme of manuscript is interesting, yet there are numerous flaws in various sections of the draft. A thorough revision is inevitable prior to publication.

This review needs to be placed in the context of previous reviews/studies for a long-standing contribution given the extensive interest in this topic. Several review articles have been recently published on similar topic. Clear discussion of previous reviews need to be made, to demonstrate how this review adds additional value to the previous reviews. The authors should provide significant new syntheses and insights at the end of each section, rather than summarizing the literature.

Authors have mentioned various sources for the preparation of plant base nanoparticles, however, failed to comparatively analyze/discuss these for practical implications. For clear understanding of the readers, a comparative discussion can be added to clarify the pros and cons of each.

Preparation of plant based polymers is itself a challenge. It will be interesting to add a flowchart to present/clarify the steps involved in the synthesis of nanoparticles.

Is there any field based evidence on the use of green-NPs… If yes, then clearly specify in a table or text.

Add the implications of the manuscript.

The challenges/limitations in NPs research/technology should be added for clarity.

The language of the manuscript needs to be improved by a native English Speaker/Professional Editor. There are numerous major/minor grammatical mistakes in the ms.

Avoid starting a sentence with an abbreviation. Define all the abbreviation used in tables in footnote.

Conclusions should summarize the key findings of ms. Avoid general statements.

Author Response

Reviewer#2

The theme of manuscript is interesting, yet there are numerous flaws in various sections of the draft. A thorough revision is inevitable prior to publication.

  • This review needs to be placed in the context of previous reviews/studies for a long-standing contribution given the extensive interest in this topic. Several review articles have been recently published on similar topic. Clear discussion of previous reviews need to be made, to demonstrate how this review adds additional value to the previous reviews. The authors should provide significant new syntheses and insights at the end of each section, rather than summarizing the literature.
  • Author’s response- As per the reviewers’ comment, we agreed that several review articles have been recently published on a similar topic, for instance, Banerjee and Roychoudhury, 2022 and Cuong et al., 2022, however, previous studies only targeted the role of only one or two NPs and focuses majorly on chemically-synthesized NPs, but our present review article gives a comprehend representation and discuss the role of various green-synthesized NPs in reprogramming plant growth and development in response to optimal as well as abiotic stress circumstances.

  • Authors have mentioned various sources for the preparation of plant base nanoparticles, however, failed to comparatively analyze/discuss these for practical implications. For clear understanding of the readers, a comparative discussion can be added to clarify the pros and cons of each.
  • Author’s response- As per the reviewers’ comment, practical implications have been added in section 6.

  • Preparation of plant based polymers is itself a challenge. It will be interesting to add a flowchart to present/clarify the steps involved in the synthesis of nanoparticles.
  • Author’s response- As per the reviewers’ comment, a flow chart has been added to the manuscript.

  • Is there any field based evidence on the use of green-NPs… If yes, then clearly specify in a table or text.
  • Author’s response- As per the reviewers’ comment, there is few field-based evidences which depicts the role of NPs in plant growth and development which has been added in table 3 (Please see the references 72, 74, 83, and 104).

  • Add the implications of the manuscript.
  • Author’s response- As per the reviewers’ comment, practical implications have been added in section 6.

  • The challenges/limitations in NPs research/technology should be added for clarity.
  • Author’s response- As per the reviewers’ comment, challenges/limitations has been added in section 6.

  • The language of the manuscript needs to be improved by a native English Speaker/Professional Editor. There are numerous major/minor grammatical mistakes in the ms.
  • Author’s response- As per the reviewers’ comment, grammatical errors have been improved in the manuscript.
  • Avoid starting a sentence with an abbreviation. Define all the abbreviation used in tables in footnote.
  • Author’s response- As per the reviewers’ comment, changes have been made in the manuscripts as per the suggestions.

  • Conclusions should summarize the key findings of ms. Avoid general statements.
  • Author’s response- As per the reviewers’ comment, the conclusion part has been rewritten.

 ***************************************************************************

Reviewer 3 Report

The review paper by authors Kumari et al. is an attempt to present the increasing usage of biologically synthesized nanoparticles to improve plant response to different abiotic stresses. In my opinion the idea behind the paper is very good because there is a lack of well-structured review papers on this subject and it would be great if one could find all the relevant information in one publication. However, even the paper is more or less well structured it can not be published in the present form due to several major and minor flaws.

Major flaws:

  1. The first one is a really poor English language, which makes a huge obstacle for reading, particularly since this is a review paper. Apart from the many mistakes in the use of the proper English grammar, the text is slovenly and sloppily written and filled with wrongly connected words and sentences without full stops. Therefore, it is mandatory that authors use the professional help of the English editing service and provide a certificate that the paper has been edited and improved!

Moreover, many sentences are starting with references number, which is very annoying for reading and should be corrected!

  1. Some parts of the manuscript are really not informative. Sentence like “Seed exudates, seed powder extract or seed endosperm extract also used for the purpose of NP synthesis“ should be avoided. In each sentence a complete and precise information should be given: which plant species was used? What type(s) of NPs were produced? What were the characteristics of the produced NPs considering their shape, concentration and stability as well as their application?
  2. There is a lack in consistency in the amount of data between different subsections of the manuscript. Namely, in “3. Green synthesized nanoparticles-mediated reprogramming of plant traits” authors gave very detailed descriptions of the studies and provided information about the type, shape and concentration of applied NPs. However, in the following subsections information is very scarce about the characteristics of applied NPs. I can understand that maybe shape of NPs is not so crucial, but the type and concentration of the employed nanoparticles definitely are extremely important. Therefore, these subsections should be supplemented with additional information.
  3. Another very serious issue is the 6th section of this review (Multi-omics approaches mediated by NPs for regulating abiotic stress tolerance). Namely, by careful check up of the studies described in this section and listed in the Table 4 I have found out that majority of them have not been performed with biologically obtained NPs. Those studies should be omitted from the Table 4 and from the text since they are not the subject of this review which should deal with bio-synthesized NPs as it is stated in the title! Therefore, Table 4 has to be carefully revised and this whole section should be carefully rewritten since it describes many studies which were not performed with green-synthesized or biologically synthesized NPs! Only the studies that used biologically synthesized NPs can be included with a clear information which plant species and/or plant parts were used to produce NPs!
  4. The similar issues is with the 7th section (Post-transcriptional regulation of stress responsive genes by nanoparticles) as well in which none of the studies discussed were performed with biologically synthesized NPS. Even worse, for one study authors state that some NP-responsive miRNAs were detected (ref 155), although in that study the effect of any NPs was not studies at all! Therefore, this whole section has to be omitted.
  5. Conclusion has to be carefully rewritten.
  6. All Tables should have the same column titles for the same data.

All minor issues are given in the pdf of the manuscript.

Author Response

Reviewer#3

The review paper by authors Kumari et al. is an attempt to present the increasing usage of biologically synthesized nanoparticles to improve plant response to different abiotic stresses. In my opinion the idea behind the paper is very good because there is a lack of well-structured review papers on this subject and it would be great if one could find all the relevant information in one publication. However, even the paper is more or less well structured it cannot be published in the present form due to several major and minor flaws.

Major flaws:

  1. The first one is a really poor English language, which makes a huge obstacle for reading, particularly since this is a review paper. Apart from the many mistakes in the use of the proper English grammar, the text is slovenly and sloppily written and filled with wrongly connected words and sentences without full stops. Therefore, it is mandatory that authors use the professional help of the English editing service and provide a certificate that the paper has been edited and improved!

 Author’s response- As per the reviewers’ comment, English language has been improved and manuscript has been certified from English editing service.

Moreover, many sentences are starting with references number, which is very annoying for reading and should be corrected!

Author’s response- As per the reviewers’ comment, in-text citations has been corrected.

  1. Some parts of the manuscript are really not informative. Sentence like “Seed exudates, seed powder extract or seed endosperm extract also used for the purpose of NP synthesis“should be avoided.

Author’s response- As per the reviewers’ comment, sentences has been removed.

In each sentence complete and precise information should be given: which plant species was used? What type(s) of NPs were produced? What were the characteristics of the produced NPs considering their shape, concentration and stability as well as their application?

Author’s response- Complete and precise information has been added as suggested by the reviewer.

  1. There is a lack in consistency in the amount of data between different subsections of the manuscript. Namely, in “3. Green synthesized nanoparticles-mediated reprogramming of plant traits” authors gave very detailed descriptions of the studies and provided information about the type, shape and concentration of applied NPs. However, in the following subsections information is very scarce about the characteristics of applied NPs. I can understand that maybe shape of NPs is not so crucial, but the type and concentration of the employed nanoparticles definitely are extremely important. Therefore, these subsections should be supplemented with additional information.

Author’s response- As per the reviewers’ comment, additional information in section 3 has been added

  1. Another very serious issue is the 6th section of this review (Multi-omics approaches mediated by NPs for regulating abiotic stress tolerance). Namely, by careful check up of the studies described in this section and listed in the Table 4 I have found out that majority of them have not been performed with biologically obtained NPs. Those studies should be omitted from the Table 4 and from the text since they are not the subject of this review which should deal with bio-synthesized NPs as it is stated in the title! Therefore, Table 4 has to be carefully revised and this whole section should be carefully rewritten since it describes many studies which were not performed with green-synthesized or biologically synthesized NPs! Only the studies that used biologically synthesized NPs can be included with a clear information which plant species and/or plant parts were used to produce NPs!

Author’s response- Section 6 and Table 4 has been omitted as the mentioned studies were not performed with green synthesized nanoparticles.

  1. The similar issues is with the 7th section (Post-transcriptional regulation of stress responsive genes by nanoparticles) as well in which none of the studies discussed were performed with biologically synthesized NPS. Even worse, for one study authors state that some NP-responsive miRNAs were detected (ref 155), although in that study the effect of any NPs was not studies at all! Therefore, this whole section has to be omitted.

Author’s response- As per the reviewers’ comment, section 7 has been omitted

  1. Conclusion has to be carefully rewritten.

Author’s response- As per the reviewers’ comment, the conclusion part has been rewritten.

  1. All Tables should have the same column titles for the same data.

Author’s response- As per the reviewers’ comment, all tables have been modified

  1. All minor issues are given in the pdf of the manuscript.

Author’s response- As per the reviewers’ comment, all minor issues have been checked and addressed

*************************************************************************************

Round 2

Reviewer 2 Report

The manuscript has been improved as per comments; I am happy to recommend acceptance.

Reviewer 3 Report

The paper has been substantially improved. Authors have corrected all issues and modified the paper according to my suggestions. The language has been professionally edited and improved. Therefore, I recommend this paper to be published.